Impact of rainfall on root water uptake in two characteristic species of coal mining subsidence areas in Northwest China

He Ruimin 1 2 3
Wei Haoyan why2020@nwafu.edu.cn 4
Lei Mingzhe 3
Niu Jiping 4
Xing Zhenguo 20039424@ceic.com 1 3 4 5
Chen Shi 4
Lei Da 1 3 4 5
Liu Gang 1 2 3
Guo Min 1 2 3
Lei Yang 1 2 3
Li Min 4
1 State Key Laboratory of Water Resource Protection and Utilization in Coal Mining , Yulin , China
2 Technology Research Institute, China Energy Shendong Coal Group Co., Ltd. , Yulin , China
3 National Energy Investment Group Co., Ltd. , Beijing , China
4 State Key Laboratory of Water Resource Protection and Utilization in Coal Mining , Beijing , China
5 National Institute of Clean-and-Low-Carbon Energy , Beijng , China
Zhang Qianwen
Electronic publication date: 2025 Oct 9
Publication date: 2025
Volume: 13
Electronic Location ID: e20158
Received 2025 Apr 4; Accepted 2025 Sep 10
Copyright: ©2025 He et al.
Copyright year: 2025
Copyright holder: He et al.
License: This is an open access article distributed under the terms of the Creative Commons Attribution License, which permits unrestricted use, distribution, reproduction and adaptation in any medium and for any purpose provided that it is properly attributed. For attribution, the original author(s), title, publication source (PeerJ) and either DOI or URL of the article must be cited.
License URL: https://creativecommons.org/licenses/by/4.0/

Keywords: Coal mining, Subsidence, Plant water uptake, Stable isotope, CrisPy model

Funding: National Energy Group Technology Project Open Fund of State Key Laboratory of Water Resource Protection and Utilization in Coal Mining NICE_RD_2025_99 This work was supported by National Energy Group Technology Project; Open Fund of State Key Laboratory of Water Resource Protection and Utilization in Coal Mining (NICE_RD_2025_99). The funders had no role in study design, data collection and analysis, decision to publish, or preparation of the manuscript.

==============================
Clarifying how plants utilize water in coal mining subsidence zones is essential for grasping plant-soil dynamics and guiding ecological rehabilitation. However, current knowledge on species-specific variations in water uptake and their adaptive responses to such subsidence remains limited. This research leveraged isotopic fingerprinting (δ2H, δ18O) alongside soil water content and root distribution to explore the root water uptake sources of two predominant species (Stipa bungeana Trin. and Artemisia desertorum Spreng.) in coal mining zones and their reactions to land subsidence triggered by coal extraction. The results indicated negligible differences in soil water content and soil water isotopic composition between subsidence and non-subsidence zones, irrespective of rainfall. Before rainfall, the water sources of the two species were unaffected by subsidence; however, after rainfall, discernible changes occurred. Plants in the subsidence area absorbed more water from the top 0–10 cm soil layer, indicating a more pronounced response to rainwater infiltration. Notably, A. desertorum, in contrast to S. bungeana, tapped into deeper soil water during arid conditions and swiftly switched to shallow soil water sources following rainfall, highlighting its adaptable water usage strategy and greater ecological resilience. The findings of this study cast new light on plant-water relationships in coal mining subsidence regions, providing essential guidance for ecological restoration and management efforts.

Introduction

As a key energy source in many nations, coal makes up over 20% of global primary energy use and is integral to industrial and social development (Kholod et al., 2020). Underground mining, the predominant method for coal extraction, offers high efficiency, safety, and minimal surface disruption (Seguel et al., 2022). However, it leads to extensive subsurface voids, causing land subsidence and altering the original environment of the Earth’s Critical Zone (Dejun, Zhengfu & Shaogang, 2016; Zhang et al., 2023). Such modifications critically reshape hydrological cycling processes by altering infiltration dynamics, surface runoff generation, land evaporation, and plant transpiration (Yang et al., 2023), leading to ecological issues like vegetation degradation and environmental deterioration in mining areas, affecting social stability and sustainable development (Li et al., 2013; Huang et al., 2022). Within water-scarce extraction zones, these challenges are particularly acute (Dai et al., 2015; Chen et al., 2023). For every stakeholder involved, reducing the ecological footprint caused by land subsidence, along with implementing effective land reclamation and vegetation restoration measures, is of paramount importance.

Root water uptake is of great significance as it acts as a vital bridge in the exchange and circulation of substances between the atmospheric, plant, and soil components (Richter & Mobley, 2009; Shao et al., 2018). Decoding the patterns of root water uptake unveils vegetation adaptive fitness strategies, particularly phytohydroregulatory feedbacks to environmental changes and such insights provide actionable frameworks for effective water management and the successful restoration of land (Gui et al., 2024; Wu et al., 2022). Studies on the plant water use strategies under various thinning measures provide new insights for forest management (Wang et al., 2022a; Wang et al., 2022b), while in-depth research on crop root water uptake offers theoretical support for precision irrigation in agriculture (Muñoz Villers et al., 2020). Furthermore, comparative analyses of the water absorption mechanisms among various types of trees in dry land offer crucial perspectives, which are highly beneficial for making informed decisions regarding the choice of plant species in the context of ecological restoration initiatives (Wang et al., 2017). Coal mining-induced subsidence brings about substantial changes to the local hydrological and soil environments, and also has a profound impact on the root systems of plants (Bi et al., 2019; Liu et al., 2024). The ecological impacts of coal mining subsidence critically impair vegetation viability and sustainable development, necessitating scientific investigations to elucidate plant-water adaptation mechanisms in subsidence-affected ecosystems. This endeavor holds immense significance as it enables a deeper understanding of the intricate relationship between plants and water resources. Moreover, it provides essential guidance for implementing effective ecological restoration strategies in subsidence areas, ultimately contributing to the sustainable recovery and rejuvenation of the local ecosystems.

Stable water isotope ratios (2H/18O) function as sensitive hydrological tracers through characteristic fractionation signatures that enable precise tracking of water cycling processes including source differentiation, transition dynamics, and hydrological mixing regimes (Dee et al., 2023; Wei et al., 2023). Employing isotopic tools to track the origin of water absorbed by plant roots requires only a small quantity of samples for analysis, thereby causing negligible harm to plants or soil while generating comparatively trustworthy results (Dawson et al., 2002; Tao, Neil & Si, 2021). Nowadays, isotopic tracing approaches have emerged as a crucial means for investigating the water uptake by plant roots and are extensively utilized in relevant research fields (Marx et al., 2022; Scandellari et al., 2024). In recent years, new models for quantifying root water uptake sources based on stable isotopes, such as PRIME (Neil, Fu & Si, 2024) and CrisPy (Fu et al., 2024), have emerged, enhancing the accuracy of isotopic tracing methods in tracking the source of water uptake by plant. Nevertheless, up to now, the number of studies that make use of stable isotopes to delve into how plant root water uptake is affected by land subsidence caused by coal mining remains relatively small. For instance, Li et al. (2013) reported alterations in the water uptake strategies of trees as a result of land subsidence. In contrast, Wei et al. (2024) proposed that sedimentation has a negligible effect on the root water absorption in herbaceous plants. Despite these contributions to the discourse on plant water use strategies in the context of coal mining subsidence, knowledge about the differences in water uptake patterns among different species and their particular reactions to subsidence is still restricted.

To bridge the existing knowledge gap, our study utilizes stable isotopes. By concurrently considering soil water content and root distribution, we aim to explore the impact of coal mining induced land subsidence on the root water uptake of two representative plant species (Stipa bungeana Trin. and Artemisia desertorum Spreng.). The specific objectives of this study are: (1) to analyze how ground sinking caused by mining activities alters moisture distribution patterns in soil layers; (2) to examine differences in water acquisition of S. bungeana and A. desertorum between subsidence and non-subsidence regions; (3) to identify the disparities in water uptake patterns between these two plant species. This research contributes new insights into the relationships between different plants and water in coal mining subsidence areas, providing a foundation and reference for ecological restoration and management in these regions.

Material and Methods

Study site

The study area positions within the transitional ecotone where Mu Us Sandy Land meets the Loess Plateau (110.15°E, 39.23°N) (Fig. 1A). This area experiences a typical arid/semiarid continental monsoon climate, marked by pronounced water deficit dynamics with annual precipitation below 400 mm contrasting sharply against evaporation rates exceeding 1,500 mm (Pei et al., 2023a). The soil profile exhibits textural homogeneity dominated by sand fractions (>90%, Fig. 1B), and has a bulk density of 1.59 g cm−3. The region’s limiting hydrological conditions support native xerophytic plant communities, primarily comprising drought-resistant grassland mosaics and scrubland vegetation (Mi et al., 2023). Among them, representative herb (S. bungeana, Fig. 1C) and shrub (A. desertorum, Fig. 1D) were selected as the research objects. S. bungeana is a perennial herb with relatively short stature and fibrous root system—its fibrous roots help it efficiently absorb shallow soil moisture, which is its key water-use adaptation to the study area’s environment. A. desertorum, by contrast, is a woody shrub with more developed aboveground biomass; it has water-use strategies like storing more water in stems and growing deeper roots to access subsoil water, making it the dominant shrub species in the region. Given the groundwater table’s position below the phreatic zone, soil moisture constitutes the sole available reservoir for vegetation moisture absorption under these hydrological constraints (Chen et al., 2022). We selected distinct flat sites in the coal mining subsidence area, one covered by S. bungeana and another by A. desertorum, with corresponding plots of the same species in the non-subsidence area. The growth characteristics of the plants in the two plots are quite similar as plant height and crown width of both species are comparable between subsidence and non-subsidence areas. The post-mining terrain has entered a geomechanical equilibrium phase, with surface fissure development ceased and displacement rates stabilized at baseline levels.

Figure 1 Field sampling site and stable isotopes of precipitation monitoring site established by Zhao et al. (2022) (A); soil texture fraction of sampling field (B); on-site photos (C, D).

Collection of rainfall, stem and soil samples

On July 1, 2022, prior to rainfall, and July 3, 2022, following rainfall, soil and plant stem samples were obtained from both the regions without subsidence and the areas affected by subsidence. Concurrently, during a rainfall event on July 2, 2022, which measured 9.6 mm and lasted 11 h, rainfall was collected using an open field rain gauge. Based on the regional long-term (1961–2020) precipitation data, the frequency of events with rainfall <10 mm exceeds 95% (Table 1). Post-collection, the rainwater was promptly transferred into 50 mL plastic bottles to prevent evaporation. Subsequently, these bottles were tightly sealed, carefully packaged, and stored in a refrigerated environment for preservation purposes. They were kept in this state until they could be subjected to isotope analysis in the laboratory.

Table 1 Cumulative frequency of rainfall.

Calculation based on long-term data from 1961 to 2020.

Rainfall amount	Cumulative frequency	
<10 mm	96.35%	
<20 mm	98.70%	
<50 mm	99.77%	
<100 mm	99.97%	
<150 mm	100.00%	

From each experimental plot, the plants that were growing healthily were chosen as the experimental subjects for sample collection. At each experimental plot, triplicate samples were gathered from three distinct sites. To conduct the analysis of plant water isotopes, samples of thick stems (with a diameter ranging from five to eight mm) were chosen as these stems were representative of the rhizome-binding part of the plants—selected for their mature vascular structure. These stems were carefully cleaned to remove any surface contaminants before being promptly transferred into pre-wash 12 mL vials (Exetainers, Labco Ltd., UK). To prevent any contamination, the vials were hermetically sealed and subsequently placed in low-temperature environment for storage, pending further isotopic determination.

Regarding the soil sampling process, at each experimental plot, three sets of parallel soil profiles were set up at various positions. These positions were close to the spots where the stem samples had been gathered. At each site, a soil profile was excavated to a depth of 100 cm, with soil being sampled at 10 cm intervals in the top 20 cm and at 20 cm intervals from 20 cm to 100 cm. The soil retrieved from each stratum was separated into two parts. One part was put into an aluminum container for the purpose of determining the gravimetric water content. This determination process required oven-drying the soil at 105 °C for a duration of 24 h. Subsequently, the gravimetric water content was converted to volumetric water content based on the measured soil bulk density of 1.59 g cm−3. Remaining soil were transferred to 100 mL polypropylene containers, hermetically sealed, and cryopreserved at 4 °C for subsequent stable isotopic determination.

Sampling and analysis for root samples

Root samples, together with the associated soil, were collected using a hand auger having an inner diameter of 85 mm. The collection was carried out at depth intervals that matched those used for the soil sampling described earlier. This sampling operation was replicated in the three locations previously mentioned.

The collected samples containing both roots and soil were then passed through 1-mm aperture matrices and rinsed with water. Then the root samples were carefully picked out using tweezers. Subsequently, the root samples were scanned at a 300 -dpi resolution. The resulting images were then processed with RhizoVision Explorer 2.0.3 (Seethepalli et al., 2021) to quantify the root length. Upon the conclusion of the scanning procedure, the root samples were transferred to an oven and subjected to drying at a temperature of 60 °C for a duration of 72 h. Following this drying phase, they were weighed to determine the dry mass. In the subsequent and final step, the root length density (RLD) was calculated by dividing the root length by the volume of the soil core. Similarly, the root mass density (RMD) was ascertained by dividing the root mass by the volume of the soil core.

Isotopic analysis of water

A vacuum condensation extraction system was utilized to extract water from both soil and stem specimens. The heating module was fully immersed within high-temperature silicone oil. Concurrently, a distinct vial designated for water collection was plunged into liquid nitrogen and then interconnected to the sample vial via a 1.5-mm stainless-steel needle. The entire system functioned under a vacuum pressure that was maintained below 1 Pa. For the soil samples, the heating temperature was set at 205 °C, while the stem samples were heated to 105 °C. The extraction of water from the soil samples was carried out for a minimum of 0.5 h, whereas the stem water extraction process lasted for 3 h. These extraction durations were carefully determined to ensure that the recovery rate of the extracted water remained within the range of 98% to 102%.

Isotopic characterization employed dual analytical platforms: Soil and meteoric water samples underwent H-O isotope determination via off-axis ICOS system (LWA-45EP; Los Gatos Research), while xylem water analysis utilized high-resolution IRMS (253 Plus; Thermo Fisher Scientific) to mitigate organic interference. All δ2H and δ18O measurements were normalized against VSMOW, with analytical precisions maintained at ±1‰ and ±0.2‰, respectively.

Although numerous studies have indicated that cryogenic vacuum extraction of stem water can introduce hydrogen isotope offsets (Chen et al., 2020; Wen et al., 2022), which is typically corrected using a source water line (Cai et al., 2024; Zhao et al., 2024), this study did not find any significant deviation between the isotopic composition of stem and soil water. Consequently, no correction was applied to the plant water data.

Data analysis

The CrisPy model, a Python script, describes root water uptake in a continuous pattern, instead of dividing the soil profile into segments (Fu & Si, 2023). The virtual and field tests illustrated that compared to MixSIAR, CrisPy generates more accurate results with smaller errors (Fu et al., 2024). Consequently, in this research, the CrisPy model was employed to determine the origin of the water absorbed by the roots. Unprocessed isotopic signatures (both δ2H and δ18O) served as primary inputs for the CrisPy model. Prior information set as non-informative prior. For more detailed information, please refer to (Fu et al., 2024).

A paired t-test was employed to assess and identify any significant differences that existed between the various plots, specifically those categorized as non-subsidence and subsidence areas: (1) t=x¯−y¯σx2+σy2

where x¯ and y¯ are the means; and σx and σy are the standard errors of variables x and y, respectively. The statistical tests were considered significant when t > 1.96 (p < 0.05) and t > 2.58 (p < 0.01), respectively.

Results

Root, soil water content and soil water isotopes profiles

The vertical distribution of the root systems of S. bungeana and A. desertorum in both subsidence and non-subsidence zones showed a decrease with increasing soil depth, with the majority of roots concentrated within the top 0–20 cm layer, accounting for over 60% of the total root length and root mass in their respective regions (Fig. 2). Differences in RLD and RMD specific to each species were detected across various soil layers. S. bungeana displayed a more developed root system in comparison to non-subsidence areas, with significant increases in RLD and RMD (p < 0.05) within the subsided areas (Figs. 2A, 2B). When it came to A. desertorum, in contrast to the non-subsidence area, the root system in the subsidence area showed only slight change, yet a significantly increase in RLD was observable in some soil layers, while RMD significantly decreases in the 0–10 cm soil layer but increases at deeper layers (Figs. 2C, 2D).

Figure 2 Vertical distribution of root length density (RLD) (A) and root mass density (RMD) (B).

Asterisks (* and **) denote a significant difference at a significance level of p < 0.05 and p < 0.01, respectively.

As shown in Fig. 3, the vertical distribution of soil water content (SWC) and the response to rainfall event for both species in non-subsidence and subsidence areas were essentially consistent, with no significant differences. Prior to rainfall, the SWC of S. bungeana and A. desertorum ranged between 1.5% and 4.0%, exhibiting a gradual increase with depth (Fig. 3). Following rainfall, the SWC within the 0–40 cm depth range for S. bungeana in each area increased due to the precipitation, whereas for A. desertorum, the depth affected by rainfall was shallower, at 0–20 cm (Fig. 3).

Figure 3 Comparison of volumetric soil water content in non-subsidence and subsidence areas before rainfall and after it.

(A) S. bungeana, (B) A. desertorum.

Similar to the pattern of soil water content variation, in both S. bungeana and A. desertorum land, the vertical distribution of soil water isotope composition and its response to rainfall were consistent in non-subsidence and subsidence areas. Prior to rainfall, the δ2H and δ18O values of soil water in S. bungeana plots were approximately between −10‰ and −40‰, and 10‰ to 0‰, respectively (Figs. 4A, 4B), while in A. desertorum land, these values are approximately between −15‰ and −50‰, and 10‰ to −5‰, respectively (Figs. 4C, 4D), with both species showing a trend of depletion from the shallow to the deep layers. After rainfall, in both S. bungeana and A. desertorum land, the soil water isotope composition in the 0–20 cm soil layer became depleted, tending towards the isotope composition of the rainfall, indicating infiltration and recharge of this soil layer by the rainfall (Fig. 4). Conversely, within the 40–100 cm depth interval, the values of δ2H and δ18O for the soil water isotopes showed a high degree of similarity to the pre-rainfall values This indicates that they were scarcely influenced by the rainfall event. In general, regardless of S. bungeana and A. desertorum cover, under dry conditions (before rainfall) and rainfall recharge conditions (after rainfall), coal mining subsidence did not cause significant changes in soil water and its isotope composition.

Figure 4 Comparing the δ2H and δ18O values of soil water before and after rain in non-subsidence and subsidence regions.

Dual isotopes plot of various water

Based on the research by Zhao et al. (2022), the local meteoric water line was defined as δ2H = 7.67 δ18O + 5.91, R2 = 0.96). Before and after rainfall, in both subsidence and non-subsidence areas, the stable isotopes of soil water were observed to be below the defined local meteoric water line, as depicted in Fig. 5. This indicates that soil water is sourced from precipitation and has experienced evaporation-induced enrichment. In the dual-isotope space, the isotopic values of stem water and soil water overlap substantially in both areas, under both pre- and post-rainfall conditions. This overlap suggests that for S. bungeana and A. desertorum, soil water serves as the primary source of water uptake (Fig. 5). Notably, prior to rainfall, the stable isotopes of stem water in S. bungeana showed greater enrichment when contrasted with that in A. desertorum (Fig. 5), highlighting distinct water uptake patterns between these species. Following rainfall, the isotopic signatures of stem and soil water in both species, which are inclined to match those of the rainfall, reveal that both the soil and the vegetation react promptly to the rainfall events (Fig. 5). However, upon comparing non-subsidence and subsidence areas, no discernible differences were detected in the isotopic composition of various water bodies (Fig. 5).

Figure 5 Characteristics of δ2H and δ18O in different water from non-subsidence and subsidence areas for S. bungeana and A. desertorum.

Local meteoric water line (LMWL) is: δ2H = 7.67 δ18O + 5.91, R2 = 0.96 (Zhao et al., 2022).

Root water uptake source

In order to conduct a quantitative assessment of water use strategies, the CrisPy model was applied to calculate the contribution proportion of water sources for S. bungeana and A. desertorum in the two zones on various sampling dates (Fig. 6). Prior to the rainfall, as the soil depth increased, the percentage of water taken up by S. bungeana showed a downward trend. In the non-subsidence area, the contribution of water absorption from different soil depths was respectively 28.4 ± 4.5% at 0∼10 cm, 21.4 cm 13.5% at 10∼20 cm, 30.1 ± 9.4% at 20∼40 cm, 17.9 ± 5.1% at 40∼60 cm, 2.1 ± 2.6% at 60∼80 cm, and 0.0 ± 1.5% at 80∼100 cm; while in the subsidence area, these values were 21.1 ± 2.5%, 33.1 ± 5.6%, 35.4 ± 10.3%, 9.5 ± 5.1%, 0.9 ± 2.3%, and 0.0 ± 0.2% for the corresponding depths (Fig. 6A). Under dry conditions (before rainfall), the water absorption source of S. bungeana mainly came from the shallow layer of 0–40 cm, which did not change with subsidence (Fig. 6A). However, A. desertorum exhibited a markedly different water uptake pattern from S. bungeana, primarily extracting moisture from the deep soil layers between 60 to 100 cm (Fig. 6C). The contributions of different soil layers to the total water uptake were as follows: 1.1 ± 0.8%, 1.1 ± 0.7%, 2.1 ± 1.7%, 13.7 ± 2.8%, 50.6 ± 8.5%, and 31.3 ± 6.2% for the non-subsided zone; and 0.1 ± 3.8%, 0.1 ± 4.4%, 0.9 ± 8.4%, 10.1 ± 12.6%, 43.1 ± 16.0%, and 45.7 ± 21.1% for the subsidence area (Fig. 6C). Similarly, the differences in plant water sources between the non-subsidence and subsidence areas were negligible.

Figure 6 Proportions of potential water sources at various depth.

The error bars denote the standard deviations of the outcomes from the CrisPy model. (A) The situation of S. bungeana before rainfall, (B) the situation of S. bungeana after rainfall, (C) the situation of A. desertorum before rainfall, and (D) the situation of A. desertorum after rainfall. Asterisks (* and **) denote a significant difference at a significance level of p < 0.05 and p < 0.01, respectively.

However, after rainfall, the water uptake patterns of both species tended to converge, with the primary water absorption layer shifting to the 0–10 cm soil layer, demonstrating a rapid response to rainfall (Figs. 6B, 6D). In the non-subsidence area, the water uptake rates of S. bungeana at different soil layers were 71.5 ± 3.9%, 0.0 ± 3.4%, 0.5 ± 1.6%, 2.7 ± 2.5%, 8.3 ± 3.1%, and 17.0 ± 6.7%, while in the subsidence area, the water uptake rates were 99.9 ± 23.2%, 0.0 ± 22.2%, 0.0 ± 1.5%, 0.0 ± 1.0%, 0.0 ± 1.2%, and 0.0 ± 1.9% (Fig. 6B). For A. desertorum, the water uptake proportions at different depths in the non-subsidence area were 47.6 ± 13.0%, 2.3 ± 2.4%, 9.5 ± 1.7%, 14.8 ± 2.7%, 16.1 ± 3.1%, and 9.7 ± 13.6%, and in the subsidence area, they were 86.2 ± 31.6%, 0.0 ± 4.1%, 0.3 ± 2.6%, 1.9 ± 3.4%, 5.7 ± 12.5%, and 6.0 ± 24.4% (Fig. 6D). Additionally, it is noteworthy that plants in the subsidence area, including S. bungeana and A. desertorum, showed a more aggressive response to rainfall, almost entirely absorbing soil water from the surface layer (Figs. 6B, 6D).

Discussion

Impact of land subsidence on soil water

Prior to rainfall, as soil depth increases, soil water content rises while the soil water isotopes become depleted (Figs. 3 and 4). This is a typical profile characteristic because the surface is subject to evaporation, returning water to the atmosphere and causing isotopic fractionation (Skrzypek et al., 2015; Xiang et al., 2021). The soil water content (Fig. 3) and stable isotope composition (Fig. 4) show a clear response to rainfall, indicating that precipitation replenishes soil water. When contrasted with the soil water that has experienced evaporative enrichment, the stable isotopes in rainfall is notably lower. Additionally, the changes in the stable isotope signature of soil water subsequent to rainfall events can be considered a reliable indicator for determining the main infiltration zone (Dee et al., 2023). When precipitation blends with soil water, the stable isotope composition of soil water, which was initially enriched, starts to become less concentrated. The peak in the isotopic signature of soil water at a 20 cm depth (Fig. 4) indicates that the main depth for precipitation-induced soil recharge is within the shallow soil layer. This conclusion is further supported by the area where the most significant alteration in soil water content occurs, as depicted in Fig. 3. This sluggish movement reflects the dominant piston-flow pattern of water supply. However, this finding contradicts the outcomes of numerous studies, which generally posit that the disruption of the vadose zone structure due to mining activities leads to the formation of fractures and fissures that facilitate increased water infiltration, thereby exhibiting significant rapid recharge and an increase in soil water storage (Zhang et al., 2022; Boumaiza et al., 2023; Zhang et al., 2023). While this could potentially be associated with the limited amount of precipitation, earlier research has demonstrated that preferential flow has a higher likelihood of taking place during heavy rainfall occurrences (Ji et al., 2021; Xiang, Evaristo & Li, 2020). Additionally, the study area is a typical sandy region, and the loose and unstructured physical properties of the sand make it more likely for the structure of the unsaturated zone damaged by settlement to recover in a short period of time (He et al., 2020; Liu et al., 2022). Consequently, within the 0–100 cm soil profile, neither soil water content nor the stable isotopic composition of soil water changes with subsidence, whether in S. bungeana or A. desertorum land. Furthermore, while the above findings stem from observations before and after a single rainfall event, in inland arid regions like the study area, rainfall mainly consists of similar small events with low intensity (Pei et al., 2023a; Zhao et al., 2025). The rainfall amount, intensity, and duration of the event observed in this study align closely with the region’s typical patterns, thus exhibiting representativeness.

Root water uptake patterns in the context of land subsidence

Although no changes in soil hydrological processes were observed between the non-subsidence and subsidence areas at present (Figs. 3 and 4), plant roots have responded to coal mining subsidence (Fig. 2). Therefore, it is still necessary to understand whether the water uptake by plant roots in the coal mining subsidence area has changed, which will aid in the restoration and management of ecosystems (Jia et al., 2024). Our results indicate that there was no significant difference in the water source for the two types of plants in the subsidence and non-subsidence areas before rainfall (Figs. 6A, 6C). A previous study by Wei et al. (2024) also reported this phenomenon. However, some other studies propose that the land subsidence resulting from coal mining expands the depth of plant water uptake (Chen et al., 2022). In this study, S. bungeana and A. desertorum, which are herbaceous and semi-herbaceous semi-shrub plants, respectively, are both lower-tier vegetation (Wang et al., 2022a; Wang et al., 2022b; Wu et al., 2021). Lower-tier plants usually show great environmental adaptability (Ward, Wiegand & Getzin, 2013). They can change their morphological structures to adapt to complex water and soil conditions, thus optimizing their water use patterns (Asbjornsen et al., 2008; Wang et al., 2017). Therefore, S. bungeana and A. desertorum may maintain their water uptake patterns by altering their root systems. The root systems in the region affected by subsidence exhibit more pronounced development compared to those in the non-affected region, as depicted in Fig. 2. This disparity could potentially serve as evidence in support of this perspective.

After rainfall, plants in different areas responded quickly to the precipitation (Figs. 6B, 6D). A substantial body of research indicates that the majority of plants quickly adjust how they take up water, giving priority to using the shallow layer soil water replenished by rainfall (Thomas, Yadav & Šimůnek, 2020; Pei et al., 2023b), as water uptake is driven by potential differences, and ample shallow soil water allows plants to extract water with lower energy expenditure (Wu et al., 2022). However, the plant water uptake patterns showed differences after rainfall, with S. bungeana and A. desertorum in the subsidence area absorbing more precipitation-recharged soil water (Figs. 6B, 6D), indicating that plants in the coal mining subsidence area have a more aggressive water use strategy. Despite the compensation in root length growth, the land subsidence caused by coal mining failed to alter the patterns of plant water absorption during dry periods. However, it did influence how plants absorbed water in response to rainfall recharge. This suggests that the adaptive changes of plants cannot fully resist the impact of coal mining subsidence, and such impacts may be long-term.

Comparison of water use strategies in the two species

Compared to the disturbances of soil water and plant root water uptake caused by coal mining subsidence, the variations in water use strategies among these two species are more pronounced. S. bungeana concentrates its water use in the 0–40 cm soil layer before rainfall (Fig. 6A), which matches the distribution of root length (Fig. 2A). When the soil water potential at different depths is similar, the source of root water uptake is primarily limited by the distribution of the root system (Yadav, Mathur & Siebel, 2009; Zhu et al., 2024). Although there is slightly more soil water at deeper layers (Fig. 3A), it does not change the dominance of the root distribution for S. bungeana. It is not until rainfall creates a sufficient water advantage in the shallow soil (Fig. 3A) that the root water uptake source for S. bungeana is controlled by soil water (Fig. 6B). However, for A. desertorum, its root water uptake during dry condition is concentrated in the deep soil layers below 60 cm, which have relatively high soil water storage (Figs. 3B and 6C), and after precipitation event, it prioritizes matching its root water uptake with the spatial distribution of the precipitation-recharged area (Figs. 3B and 6D), indicating that A. desertorum is more sensitive to soil moisture. The differential water absorption strategies of A. desertorum and S. bungeana may correlate with their water usage, where the greater canopy area and water consumption of A. desertorum necessitate deeper water extraction to fulfill its transpiration requirements. Moreover, this ecological plasticity, switching between shallow and deep soil water, has also been observed in species such as sea buckthorn, tamarisk, and black locust (Chen et al., 2023; Wu et al., 2019). The switching of plant root water uptake between different soil layers is attributed to the dimorphic root system (Grossiord et al., 2017). Moreover, numerous studies have shown that absorbing deep soil water helps alleviate water stress, maintain normal plant water metabolism, and resist drought (Li et al., 2021a; Li et al., 2021b; Kühnhammer et al., 2023). Therefore, under similar environment, A. desertorum exhibits greater plasticity in root water uptake compared to S. bungeana.

Insights for ecological restoration in regions with land subsidence

The restoration of ecosystems degraded by ground subsidence in coal-mining regions is a global concern, with vegetation restoration being a key priority (Nuttle et al., 2017; Du et al., 2021). As a result, comprehending the soil hydrological processes and the characteristics of plant water uptake in ground-subsidence-influenced regions is crucial in providing evidence for the restoration and management of ecosystems. In this study, occurrence of subsidence did not significantly alter soil water and its isotopic composition before or after rainfall (Figs. 3 and 4), nor did it markedly change the uptake location of S. bungeana and A. desertorum under dry conditions, but plants in the subsidence area responded more aggressively to rainfall recharge (Fig. 6). This underscores that plant-soil interactions may still exhibit responses even when soil itself is not significantly affected by subsidence. Therefore, although plants exhibit resilience to the adverse effects of coal mining-induced ground subsidence, some measures may still be needed for plants in the subsidence area, such as soil improvement and the cultivation of stress-tolerant species (Bi et al., 2019; Ding et al., 2025). At the same time, it must be acknowledged that this study has limitations; soil properties vary greatly in different regions (Tong et al., 2024), and the generalizability of findings from windy desert area may be influenced by changes in soil characteristics. Moreover, our results reveal that A. desertorum has better ecological plasticity compared to S. bungeana, which is beneficial for plant survival in adverse environments, especially under the backdrop of climate change, where plants face frequent droughts and high temperatures (Li et al., 2021a; Li et al., 2021b; Madakumbura et al., 2021; Yang, Yang & Xia, 2021). Therefore, when reconstructing vegetation, A. desertorum could be chosen to cope with the continuously changing and harsh environmental conditions.

Conclusions

In this study, we employed stable water isotopes in combination with soil water content and root distribution to investigate the root water uptake sources of two typical species growing in coal mining subsidence areas in northwest China and their responses to ground subsidence induced by coal mining activities. Before and after rainfall events, negligible differences in soil water content and stable isotopes (δ2H and δ18O) were detected between non-subsidence regions and subsidence areas, owing to the insensitivity of sandy soil to structural changes in the vadose zone. Prior to rainfall, the water sources absorbed by plants in the two regions were consistent, but slight variations were noted post-rainfall, with plants in the subsidence area almost entirely absorbing from the 0–10 cm soil layer, demonstrating a more aggressive response to rainfall recharge. Plants can undergo adaptive changes in their root systems, resulting in minimal differences in water uptake patterns subsidence and non-subsidence regions. Furthermore, compared to S. bungeana, A. desertorum showed higher ecological flexibility in water use: it utilized deeper soil water during dry conditions and switched to shallow soil water after rainfall, with this dual strategy directly correlating to its adaptive adjustment to moisture fluctuations. This study presents innovative viewpoints regarding the interplay between plants and water in coal mining subsidence areas, thereby providing a fundamental underpinning and valuable reference for the design and execution of ecological restoration efforts and management approaches in such areas.

Supplemental Information

Supplemental Information 1 Data

Root, soil water content, and water isotopes of stem and soil in coal mining subsidence and non-subsidence areas.

Authors thank the technical help from Jingjing Jin, Institute of Water-saving Agriculture in Arid Areas of China, Northwest A&F University.

Additional Information and Declarations

Competing Interests

Author Contributions

Data Availability

Ruimin He, Gang Liu, Min Guo and Yang Lei are employed by China Energy Shendong Coal Group Co., Ltd. Ruimin He, Mingzhe Lei, Zhenguo Xing, Da Lei, Gang Liu, Min Guo and Yang Lei are employed by National Energy Investment Group Co., Ltd.

Ruimin He performed the experiments, analyzed the data, prepared figures and/or tables, authored or reviewed drafts of the article, and approved the final draft.

Haoyan Wei conceived and designed the experiments, performed the experiments, authored or reviewed drafts of the article, and approved the final draft.

Mingzhe Lei performed the experiments, prepared figures and/or tables, and approved the final draft.

Jiping Niu analyzed the data, prepared figures and/or tables, and approved the final draft.

Zhenguo Xing conceived and designed the experiments, analyzed the data, authored or reviewed drafts of the article, and approved the final draft.

Shi Chen performed the experiments, authored or reviewed drafts of the article, and approved the final draft.

Da Lei performed the experiments, prepared figures and/or tables, and approved the final draft.

Gang Liu analyzed the data, prepared figures and/or tables, and approved the final draft.

Min Guo conceived and designed the experiments, prepared figures and/or tables, and approved the final draft.

Yang Lei analyzed the data, authored or reviewed drafts of the article, and approved the final draft.

Min Li conceived and designed the experiments, performed the experiments, authored or reviewed drafts of the article, and approved the final draft.

The following information was supplied regarding data availability:

The raw measurements are available in the Supplemental File.

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
