# Peer review of "Impact of rainfall on root water uptake in two characteristic species of coal mining subsidence areas in Northwest China"

_PeerJ, doi:10.7717/peerj.20158_

## Round 0.1 · original submission · Major Revisions

Please take special note of the advice from Reviewer 1 regarding the need for additional data covering more rainfall events.

Reviewer 1 ·

Basic reporting

The manuscript investigates root water uptake in response to a rainfall event using stable isotope analysis and associated soil/root measurements. While the research topic is relevant and valuable in the context of plant–soil–water interactions, the manuscript currently has several methodological and presentation issues that limit its suitability for publication.

Experimental design

The main issues that limit its suitability for publication, including:
1. Limited Rainfall Observations
The study is based on a single rainfall event effect on root water uptake, which is insufficient to account for the variability in plant and soil responses. Root water uptake dynamics can be strongly influenced by factors such as rainfall intensity, duration, and antecedent moisture conditions. A more robust analysis would require observations over multiple rainfall events (ideally at least 3–5) to validate the observed patterns and support broader conclusions. Additionally, manuscript also did not provide the duration of rainfall which also important for assessing the effect on soil moisture.

2. Inconsistency in Water Content Reporting
The authors describe the measurement of gravimetric water content (GWC) in the methods section. However, Figure 3 present volumetric water content (VWC) without any explanation in methodology of how this was derived. This creates confusion and suggests either a misunderstanding of the two terms or an omission in the description of data processing. A clear explanation of the conversion from GWC to VWC is necessary to ensure transparency and reproducibility.

Validity of the findings

The conclusion is well-stated overall, but the results related to Objective #2 have not yet been fully addressed.

Additional comments

Additional comment for improve the manuscript:
1. The manuscript compares root water uptake between A. desertorum and S. bungeana, but provides limited background on the ecological and physiological characteristics of these species. For the reader to understand why differences in water uptake might occur, it is essential to include basic information on traits such as root morphology, drought tolerance, water-use strategy (e.g., deep vs. shallow rooting), and ecological roles in the study area. Additionally, in Figure 1 the authors could also provide the photo of study area that consist of the two types of plant species with marked those species so it will insight the physical characteristic of the plant species.

2. It is advisable to avoid using personal pronouns (e.g., 'we,' 'our') in the manuscript. For example, the sentence 'We leveraged isotropic fingerprint...' could be revised to the third person, such as 'This research leveraged isotropic fingerprint..".

3. In line 118, the authors state that “The growth characteristics of the plants the two plots are quite similar”. The author should provide what parameters or method the author used to specify it.

4. The author should provide the duration for collecting the rainwater during rainfall.

5. In line 132-134 the author state "To conduct the analysis of plant water isotopes, samples of thick stems were chosen. These thick stems were representative of the rhizome-binding part of the plants". The author should specify dimension of stems thickness and what is the characteristic to choose the thickness.

·

Basic reporting

The text was clear, unambiguous and professional. It was written in English. The article included a sufficient introduction and background information. Relevant prior literature was referenced appropriately. The raw data was prepared professionally and clearly. The study is innovative, and all the results are relevant to the hypothesis.


My suggestion to improve the text:
In lines 24-25 consider providing the full scientific name of Artemisia desertorum and Stipa bungeana, including the author citation, following the standard botanical nomenclature format (e.g., Lepidium sativum L.). It can be written only once in the abstract and at the beginning of the introduction, e.g. at line 95. Consider doing the same action at line 112.

Consider putting only one citation for both sentences in lines 55 and 57 if the articles are the same.
At line 369, instead of "top priority", use "key priority". At line 370, instead of "procedures", use "processes".

In lines 390–392, it would be better to add information that answers the question - Why are "negative differences" in soil water content and isotopes considered "insignificant"?

Experimental design

The experimental design was well-prepared, and the submission clearly defined the hypothesis, which was relevant to the study. The lack of knowledge is acknowledged, which is why this study is innovative. The methods are described with sufficient information.

My suggestion to improve the text:
I appreciate the work put into the graph preparation, but I have a comment and suggestion: in figure 6d in variant “after rainfall A. desertorum”, the standard deviations of subsidence and non-subsidence groups at 100 cm depth are too high. The same appears at 6b at 20 cm depth. Reconsider rearranging the graph to show the results in better way.

Validity of the findings

Validity of the findings: The results are relevant and innovative. The conclusions are clear and synthetic. I recommend reading my suggestions on what to add to make them easier for a reader to understand.

My suggestion to improve the text:
Suggestion to lines 397-399: reconsider putting a note about that A. desertorum exhibits "higher ecological flexibility" in the context of water use, to bring a better impact and detail the specific correlations in that topic.

Reviewer 3 ·

Basic reporting

The article reads well but would benefit from further editing for clarity and the removal of redundant sentences. Especially in the introduction and method section

Experimental design

The experimental design is replicable but would benefit from clearer explanation and more detailed descriptions.

Validity of the findings

The figures are well presented. However, the author could consider reducing the length of the results section and placing more emphasis on explaining the significance of the findings and their implications for plant–soil dynamics.

Additional comments

The authors have done a good job on the manuscript and are encouraged to consider the comments above. Specifically, I recommend correcting minor grammatical errors, shortening overly long phrases for clarity, and using appropriate isotopic notation (e.g., ²H and ¹⁸O).

---

## Round 0.2 · Minor Revisions

Please address comments from Reviewer 4, thanks!

Reviewer 1 ·

Basic reporting

The authors have revised the manuscript based on my comments. However, comment no 1 regarding adding more data, the author still cannot provide it due to objective difficulties in practice.

Actually, it will be a more solid conclusion if they can add more data, but the authors already stated that the measured rainfall data can be representative of the area.

Experimental design

-

Validity of the findings

-

Reviewer 3 ·

Basic reporting

-

Experimental design

-

Validity of the findings

-

·

Basic reporting

It would be helpful to include each species’ water-use strategy and adaptations in the Introduction or in Section 2.1 to better support the interpretation of their uptake differences.

Experimental design

Please consider adding a small chart or table with historical data showing how this 9.6 mm event compares to typical rainfall in the area. If that data isn't available, might can adding a brief sentence to the Discussion about this limitation would also be a good solution.

Validity of the findings

The data are well presented. Please italicize all Latin names consistently throughout the text and figures, particularly in Figure 6 (d).

---

## Round 0.3 · accepted · Accept

Thank you for your time and effort in improving the manuscript. I am pleased to inform you that the revised manuscript now meets our standards for publication.

·

Basic reporting

No comment.

Experimental design

No comment.

Validity of the findings

No comment.